# PTEN Mediates the Silencing of Unintegrated HIV-1 DNA

**DOI:** 10.3390/v16020291

**Published:** 2024-02-14

**Authors:** An Thanh Phan, Yiping Zhu

**Affiliations:** Department of Microbiology and Immunology, University of Rochester Medical Center, Rochester, NY 14642, USA; an_phan@urmc.rochester.edu

**Keywords:** PTEN, HIV-1, unintegrated viral DNA, silencing

## Abstract

The integration of viral DNA into a host genome is an important step in HIV-1 replication. However, due to the high failure rate of integration, the majority of viral DNA exists in an unintegrated state during HIV-1 infection. In contrast to the robust expression from integrated viral DNA, unintegrated HIV-1 DNA is very poorly transcribed in infected cells, but the molecular machinery responsible for the silencing of unintegrated HIV-1 DNA remains poorly characterized. In this study, we sought to characterize new host factors for the inhibition of expression from unintegrated HIV-1 DNA. A genome-wide CRISPR-Cas9 knockout screening revealed the essential role of phosphatase and tensin homolog (PTEN) in the silencing of unintegrated HIV-1 DNA. PTEN’s phosphatase activity negatively regulates the PI3K-Akt pathway to inhibit the transcription from unintegrated HIV-1 DNA. The knockout (KO) of PTEN or inhibition of PTEN’s phosphatase activity by point mutagenesis activates Akt by phosphorylation and enhances the transcription from unintegrated HIV-1 DNA. Inhibition of the PI3K-Akt pathway by Akt inhibitor in PTEN-KO cells restores the silencing of unintegrated HIV-1 DNA. Transcriptional factors (NF-κB, Sp1, and AP-1) are important for the activation of unintegrated HIV-1 DNA in PTEN-KO cells. Finally, the knockout of PTEN increases the levels of active epigenetic marks (H3ac and H3K4me3) and the recruitment of PolII on unintegrated HIV-1 DNA chromatin. Our experiments reveal that PTEN targets transcription factors (NF-κB, Sp1, and AP-1) by negatively regulating the PI3K-Akt pathway to promote the silencing of unintegrated HIV-1 DNA.

## 1. Introduction

Retroviral infection begins with the reverse transcription of the RNA genome in the cytoplasm to form linear double-stranded DNA that will soon be delivered into the nucleus [1]. A portion of this linear DNA gives rise to two circular DNA forms: homologous recombination between Long Terminal Repeat (LTR) sequences at the ends of the linear DNA gives rise to 1-LTR circles, and ligation of the termini of the linear DNA by the non-homologous end joining (NHEJ) DNA repair pathway gives rise to 2-LTR circles [2,3]. In a crucial step in the retroviral life cycle, the linear DNA is inserted into the host genome to form the integrated provirus.

During the natural infection of HIV-1, the vast majority of viral DNA exists in an unintegrated state [4,5,6,7]. The unintegrated HIV-1 DNA molecules are very poorly transcribed in infected cells, in contrast to the robust expression from integrated viral DNA [8,9]. Unintegrated HIV-1 DNA is loaded with histones upon entry into the nucleus, and histones acquire repressive epigenetic marks (H3K9me3, absence of H3ac and H3K4me3) characteristic of silent chromatin [10,11]. The integrated HIV-1 DNA adopts a chromatin structure with two nucleosomes (Nuc0 and Nuc1) and a DNase hypersensitive site (DHS, characteristic of open chromatin) in the LTR region [12]. In contrast, the DHS in the LTR region of unintegrated HIV-1 DNA is covered by an additional nucleosome (NucDHS), resulting in a compact chromatin structure on unintegrated viral DNA [12]. All these observations suggest that the unintegrated HIV-1 DNA is silenced by epigenetic mechanisms. However, the host machineries and mechanisms responsible for the silencing of unintegrated HIV-1 DNA are still not well understood. 

A number of host factors have been reported to mediate the silencing of unintegrated HIV-1 DNA. NP220 binds to unintegrated HIV-1 DNA and promotes the deacetylation and silencing of the viral chromatin of unintegrated HIV-1 DNA [13]. Histone chaperones CHAF1A/B mediate the silencing of unintegrated HIV-1 DNAs in early infection with an unknown mechanism [14]. The SMC5/6 complex binds unintegrated HIV-1 DNA and promotes the silencing of unintegrated HIV-1 DNA by either the compaction or SUMOylation of viral chromatin [15,16], both of which are associated with transcriptional repression. POLE3 maintains the heterochromatin structure on unintegrated HIV-1DNA, and thus prevents the recruitment of RNAPII to HIV-1 LTR promoter [17]. The inactivation of transcription factors acting on HIV-1 LTR also represses the transcription from unintegrated HIV-1 DNA. HTLV-1 Tax protein can rescue the expression from unintegrated HIV-1 DNA by the activation of NF-κB [18].

Here, we performed a genome-wide CRISPR knockout screening and identified that PTEN is essential for the silencing of unintegrated HIV-1 DNA. In this report, we reveal that PTEN acts on transcription factors and histone modifications to mediate the silencing of unintegrated HIV-1 DNA.

## 2. Results

### 2.1. Genome-Wide CRISPR Knockout Screening for Host Factors Responsible for the Silencing of Unintegrated HIV-1 DNA

To identify new host factors responsible for the silencing of unintegrated HIV-1 DNA, we performed a genome-wide CRISPR-Cas9 knockout screening, selecting for knockout of host genes that would relieve the silencing of unintegrated HIV-1 DNA (Figure 1A). HeLa cells stably expressing Cas9 (HeLa-Cas9) were transduced with a CRISPR sgRNA expression library (4 sgRNAs per gene, targeting 19,114 genes). The resulting knockout collection of HeLa cells was infected with integrase-deficient (IN^D64A^) HIV-GFP reporter virus and the 5% brightest GFP-positive cells were sorted out. After culturing for 14 days to allow decay of the GFP, the infection and sorting procedure was repeated, and the sorted cells were expanded for analysis. In parallel, the knockout pool of HeLa cells without infection and sorting were also cultured and expanded. The abundance of sgRNAs in sorted and unsorted cells was assessed by deep sequencing. The Model-based Analysis of Genome-wide CRISPR/Cas9 Knockout (MAGeCK) method [19] was used to analyze the sgRNA sequencing results, which ranks candidate genes essential for the silencing based on a modified robust ranking aggregation (α-RRA) algorithm (Figure 1B and Appendix A). PTEN was ranked the number 1 hit from our screening.

To validate the function of PTEN in the silencing of unintegrated HIV-1 DNA, we knockout PTEN in HeLa cells. We generate two independent knockout cell lines to avoid possible clonal variance. The knockout of PTEN was verified by Western blot (Figure 1C). We then infected parental (WT) and PTEN knockout (PTEN KO #1 and #2) HeLa cells with integrase-deficient (IN^D64A^) HIV-Luc reporter virus. The knockout of PTEN increased the expression of luciferase from unintegrated HIV-1 DNA by ~4-fold (Figure 1C), indicating PTEN can mediate the silencing of unintegrated HIV-1 DNA. The increase in viral gene expression was not due to the change of the production of viral DNA, because the amount of total viral DNA in PTEN-KO cells was comparable to that in WT HeLa cells (Figure 1D). The knockout of PTEN did not change the levels of 2-LTR circular viral DNA either (Figure 1F), indicating PTEN had no effect on the nuclear entry of viral DNA. To further confirm the function of PTEN in HIV-1-targeting cells, we infected CD4+ monocyte cell line U937 with integrase-deficient (IN^D64A^) HIV-Luc reporter virus and then treated infected cells with PTEN inhibitor Disulfiram. The inhibition of PTEN in U937 cells increased the expression of luciferase from unintegrated HIV-1 DNA (Figure 1F). All these results indicated that PTEN mediated the silencing of unintegrated HIV-1 DNA.

### 2.2. PTEN’s Lipid Phosphatase Activity Mediates the Silencing of Unintegrated HIV-1 DNA by the Deactivation of Akt

PTEN primarily functions as a phosphatase for lipid, mainly PIP3 [20,21,22]. PIP3 induces phosphorylation and activates the serine/threonine kinase Akt. PTEN’s lipid phosphatase activity dephosphorylates PIP3 to PIP2 and thus induces the dephosphorylation and deactivation of Akt [23]. In addition to lipid phosphatase activity, PTEN also bears protein phosphatase activity. The G129E mutation abrogates PTEN’s lipid but not protein phosphatase activity [24]. PTEN Y16C and Y138L mutants display lipid phosphatase activity but lack protein phosphatase activity [25].

We constructed stable cell lines expressing wild-type (WT) and mutant forms (G129E, Y138L, Y16C) of PTEN in PTEN-KO HeLa cells. Both WT and mutant forms of PTEN were expressed well in PTEN-KO cells (Figure 2A). In the PTEN-KO cells, Akt was activated and displayed a high level of phosphorylation. The expression of lipid phosphatase-active PTEN (WT, Y138L, and Y16C) in PTEN-KO cells inhibited the phosphorylation of Akt, but lipid phosphatase-inactive PTEN (G129E) failed to do so (Figure 2A). We infected these cell lines with integrase-deficient (IN^D64A^) HIV-Luc reporter virus. The expression of WT, Y138L, and Y16C PTEN, but not G129E PTEN, restored the silencing of unintegrated HIV-1 DNA (Figure 2B). All these results suggest that the lipid phosphatase activity is essential for the PTEN-mediated silencing of unintegrated HIV-1 DNA.

To further confirm the importance of Akt in PTEN-mediated silencing, we infected parental (WT) and PTEN-KO HeLa cells with integrase-deficient (IN^D64A^) HIV-Luc reporter virus, treated infected cells with an allosteric and highly selective pan-AKT inhibitor MK2206 at two concentrations (2.5 μM and 5 μM) for 24 h, and assayed the luciferase expression from unintegrated HIV-1 DNA. We observed a reduction in PTEN levels in WT HeLa cells treated with MK2206, which may be due to the cytotoxic activity of MK2206 [26]. WT HeLa cells displayed a basal level of Akt phosphorylation, PTEN-KO cells displayed a high level of Akt phosphorylation, and MK2206 treatment completely abolished Akt phosphorylation in both WT and PTEN-KO cells (Figure 2C). In WT HeLa cells, the unintegrated HIV-1 DNA was silenced, and MK2206 treatment had little effect on the expression from unintegrated HIV-1 DNA (Figure 2D). The knockout of PTEN activated the expression from unintegrated HIV-1 DNA, and MK2206 treatment reduced the expression of unintegrated HIV-1 DNA back to the level in WT HeLa cells (Figure 2D). All these results suggest that Akt is important in the PTEN-mediated silencing of unintegrated HIV-1 DNA.

### 2.3. Transcription Factors (NF-κB, Sp1, and AP-1) Are Required for the Activation of Unintegrated HIV-1 DNA in PTEN-KO Cells

Cellular transcription factors (NF-κB, Sp1, and AP-1) are recruited to the 5′LTR region to initiate the transcription of HIV-1 DNA [27]. All these transcription factors are associated with the activation of the Akt pathway. Activation of the Akt pathway upregulates the transcriptional activity of the nuclear factor-κB (NF-κB) by inducing the phosphorylation and subsequent degradation of inhibitor of κB (I-κB) [28]. Activated Akt also phosphorylates Sp1 and Activator protein 1 (AP-1), and thus enhances the transcription activity of Sp1 and AP-1 [29,30].

To test the importance of these transcription factors (NF-κB, Sp1, and AP-1) in the activation of unintegrated HIV-1 DNA in PTEN-KO cells, we individually knocked down the expression of RELA (a subunit of the NF-κB), Sp1, and JUN (a member of AP-1) in HeLa cells using siRNAs (Figure 3A), and then challenged each culture by infection with integrase-deficient (IN^D64A^) HIV-Luc reporter virus. The knockdown of these transcription factors reduced the activation of unintegrated HIV-1 DNA in PTEN-KO cells (Figure 3B). All these results suggest that NF-κB, Sp1, and AP-1 are activated in PTEN-KO cells to initiate the transcription from unintegrated HIV-1 DNA.

### 2.4. The Knockout of PTEN Promotes the Deposition of Active Epigenetic Marks on Unintegrated HIV-1 DNA

Unintegrated HIV-1 DNA is loaded with histones upon entry into the nucleus and the viral chromatin lacks active epigenetic marks [11,15,16]. To assess the consequences of PTEN KO for histone modifications on unintegrated HIV-1 DNA, we performed ChIP assays with antibodies specific for active epigenetic marks (H3K9ac and H3K4me3). The knockout of PTEN had no effect on the loading of histones on unintegrated HIV-1 DNA (Figure 4A). The knockout of PTEN increased the levels of H3K9ac and H3K4me3 on unintegrated HIV-1 DNA (Figure 4B,C), indicating that the knockout of PTEN promotes the deposition of active epigenetic marks to enhance the transcription from unintegrated HIV-1 DNA. Consistent with the enhanced transcription, the knockout of PTEN increased the recruitment of PolII on the LTR promoter region of unintegrated HIV-1 DNA (Figure 4D).

## 3. Discussion

In this report, through genome-wide CRISPR knockout screening in HeLa cells, we identified PTEN as a key host factor that is required for the silencing of unintegrated HIV-1 DNA. Mechanistically, PTEN blocks the transcription from unintegrated HIV-1 DNA by inhibition of the PI3K-Akt signaling pathway. Recently, another two CRISPR knockout screenings in the T cell line have identified SMC5/6 as essential host factors for the silencing of unintegrated HIV-1 DNA [15,16]. The SMC5/6 complex promotes the compaction and SUMOylation of unintegrated viral chromatin, and thus blocks the transcription from unintegrated HIV-1 DNA. The other two CRISPR screenings did not identify PTEN because these two screenings used T cell lines (Jurkat or CEM-SS) which are PTEN expression-negative [31,32]. Our screening did not identify the SMC5/6 complex, suggesting the silencing of unintegrated HIV-1 DNA is mediated by a distinct set of host factors in different cell lines. 

PTEN is expressed in CD4 T cells [32]—the primary targets of HIV-1 infection. We do not yet know whether PTEN acts independently or corporately with other host factors to mediate the silencing of unintegrated HIV-1 DNA. Host cell transcription factors (NF-κB, Sp1, and AP-1) that are required for the transcriptional initiation of HIV-1 DNA are under the regulation by PTEN via the Akt pathway. Our results also showed that the knockout of PTEN increased the active epigenetic marks on the unintegrated HIV-1 DNA chromatin. So, it is likely that PTEN inhibits the transcriptional initiation of viral DNA and also acts synergically with other host factors (such as the SMC5/6 chromatin remolding complex and POLE3) to induce the formation of heterochromatin on unintegrated HIV-1 DNA. 

During the natural infection of HIV-1, the vast majority of viral DNA exists in an unintegrated state [4,5,6,7]. The activation of viral gene expression from unintegrated viral DNA might benefit HIV-1 infection. Reversal of the silencing of unintegrated viral DNA accelerates the replication and dissemination of HIV-1 in culture cells [13]. The transcription from unintegrated HIV-1 DNA can be activated in many different ways. HIV-1 accessory protein Vpr is incorporated into virions, and virally imported Vpr can enhance viral gene expression from unintegrated HIV-1 DNA [33]. Transcriptional regulators of other viruses, such as HTLV-1 Tax, can prevent the silencing of unintegrated HIV-1 DNA. HTLV-1 Tax activates the NF-κB pathway and rescues viral gene expression from the unintegrated HIV-1 DNA at levels sufficient to produce a progeny virus, which allows the robust replication of HIV-1 in the absence of viral DNA integration [18]. Consequently, it is reasonable to speculate that integrase inhibitors may be less effective in the treatment of HIV-1/HTLV-1 coinfected individuals. Short-chain fatty acids produced by gut microbial flora act as histone deacetylase inhibitors (HDACis) to activate viral gene expression from unintegrated HIV-1 DNA, which may account for stimulated HIV-1 replication in the gut and explain the high efficiency of HIV-1 transmission to intestinal T cells [10]. Our data suggest that enhancing PTEN activity might serve as a new strategy to inhibit HIV-1 infection.

Unintegrated HIV-1 DNA (especially the circular forms) can persist in slowly or non-dividing cells (e.g., resting CD4 T cells and macrophages) over long periods [34,35,36]. The persistence and silencing of unintegrated HIV-1 DNA suggest these viral DNA molecules may contribute to HIV-1 latency—the major obstacle to an HIV-1 cure [37,38,39]. In fact, unintegrated viral DNA is the predominant form of HIV-1 DNA that persists in the brain microglia of patients under antiretroviral therapy [40]. Disulfiram, an FDA-approved drug for the treatment of alcoholism, can reduce the expression of PTEN [31]. Indeed, disulfiram treatment can reactivate latent HIV-1 DNA primary CD4 T cells [41]. A previous study also showed that Akt is required for HIV-1 production from long-living virus-infected macrophages [42], suggesting the inhibition of PTEN, which activates of Akt, may also reactivate the integrated HIV-1 DNA in latently infected macrophage cells. Our identification of PTEN as a host factor essential for the silencing of unintegrated HIV-1 DNA suggests that the inhibition of PTEN (e.g., by disulfiram) might promote the reactivation of latent HIV-1 DNA and allow for ultimate virus clearance.

## 4. Material and Method

### 4.1. Cell Lines

Human HeLa cell line (ATCC, CCL-2) and human Lenti-X 293T cell line (Clontech, 632180) were maintained at 37 °C and 5% CO_2_ in DMEM supplemented with 10% inactivated fetal bovine serum. U937 cells (ATCC, CRL-1593.2) were cultured at 37 °C and 5% CO_2_ in RPMI 1640 supplemented with 10% inactivated fetal bovine serum.

### 4.2. DNA Construction

HIV-1 vector pNL4.3-Luc (HIV-Luc) IN^D64A^ (replication-defective, integrase-deficient, carrying firefly luciferase reporter gene) was described previously [13]. HIV-1 vector pNL4.3-GFP (HIV-GFP) IN^D64A^ was constructed by replacing the luciferase coding sequence (CDS) in pNL4.3-Luc IN^D64A^ with GFP CDS. pMD2.G expresses the vesicular stomatitis virus (VSV) envelope glycoprotein. Coding sequences for PTEN (WT or mutations) were cloned into pLvx-EF1-IRES-Neo [13].

### 4.3. DNA Transfection, Virus Package, and Infection

All the DNA transfections were performed using lipofectamine 2000 (Invitrogen) following the manufacturer’s protocol.

To package VSV g pseudotyped HIV-1 reporter virus, pNL4.3-Luc IN^D64A^ or pNL4.3-GFP IN^D64A^, together with pMD2.G, were transfected into 293T cells. To package lentiviral vector (pLvx-EF1-IRES-Neo)-based VSV g pseudotyped viruses, viral vectors together with pCMVdeltaR8.2 (expressing HIV-1 Gag and Gag-Pol) and pMD2.G were transfected into 293T cells. Forty hours after transfection, supernatants were collected and filtered through a 45 µm membrane to produce virus preparations.

Unless otherwise indicated, viruses were 3-fold-diluted with cell culture medium containing 20 mM HEPES (pH 7.5) and 4 µg/mL polybrene. Three hours after infection, the cell culture medium was changed.

### 4.4. Genome-Wide CRISPR-Cas9 Screen

The Human CRISPR Knockout Pooled Library (Brunello, 2 vector system) was obtained from Addgene. The CRISPR sgRNA library virus was packaged by transfecting 293T cells with library DNA, HIV-1 Gag-Pol-expressing plasmid pCMVdeltaR8.2, and pMD2.G. HeLa cells were transduced with lentiCas9-Blast virus, and two days after transduction, cells were selected in 5 µg/mL blasticidin for two weeks to obtain pooled HeLa-Cas9 cells. A total of 10^8^ Hela-Cas9 cells were transduced with CRISPR sgRNA library virus at M.O.I. ~ 0.3. Two days after transduction, cells were selected in medium containing 1 µg/mL puromycin for two weeks to obtain the collection of pooled HeLa knockout cells, and cells were cultured in medium containing 5 µg/mL blasticidin and 1 µg/mL puromycin during the whole process of screening. A total of 2 × 10^7^ pooled KO HeLa cells were infected with integrase-deficient (IN^D64A^) HIV-GFP virus (3-fold dilution), and 5% of the brightest GFP cells were sorted by FACS. The sorted cells were expanded for 2 weeks and then the above infection/sorting procedure was performed for the second time. Genomic DNA was extracted from the resulting selected cells and the control cells (transduced with the sgRNA library, cultured in parallel but without infection/sorting). The abundance of sgRNAs in the control cells and the sorted cells was analyzed by PCR followed by next-generation sequencing (NGS). The PCR to amplify the sgRNA was performed as described in step 32–33 of the previously published protocol [43]. The sequencing results were analyzed using MAGeCKFlute [44]. The counts of sgRNAs in each sample were calculated using “mageck count”, and the screen hits were identified using MAGeCK (Version 0.5.9).

### 4.5. Luciferase Assay

Luciferase activity was assayed 40 h after infection, using the Luciferase Assay System (Promega, Madison, WI, USA).

### 4.6. Chromatin Immunoprecipitation (ChIP)

A total of 2 × 10^6^ cells were seeded in 10 cm dishes and infected with VSV g pseudotyped, integrase-deficient HIV-Luc virus for two days. The virus used for infection was pretreated with 5 U/mL DNase (Promega, M6101) supplemented with 10 mM MgCl_2_ at 37 °C for 1 h to remove any residual plasmid DNA. Cells were crosslinked in 1% formaldehyde for 10 min, quenched in 0.125 M glycine for 5 min, and lysed in 1 mL of ChIP lysis buffer (50 mM Tris-HCl pH 8.0, 1% SDS, 10 mM EDTA, complemented with Protease inhibitor cocktail). Cell lysates were sonicated using Branson 150 Digital Sonifier (30% power amplitude, 30” × 8 times, on ice for 60” between each sonication) to produce an average chromatin fragment size of 200–800 base pairs and centrifuged at 13,000 rpm at 4 °C for 20 min. The supernatant comprising ~50 μg of sonicated chromatin was then immunoprecipitated overnight using 5 µg of specific antibodies in ChIP dilution buffer (10 mM Tris-HCl pH 8.0, 1% Triton X-100, 0.1% SDS, 150 mM NaCl, 2 mM EDTA). The next day, 25 μL of Dynabeads (12.5 µL protein A + 12.5 µL protein G) was added and incubated for an additional 2 h. The beads were washed twice each in ChIP low-salt buffer (20 mM Tris-HCl pH 8.0, 1% Triton X-100, 0.1% SDS, 150 mM NaCl, 2 mM EDTA), ChIP high-salt buffer (20 mM Tris-HCl pH 8.0, 1% Triton X-100, 0.1% SDS, 500 mM NaCl, 2 mM EDTA), ChIP LiCl buffer (10 mM Tris-HCl pH 8.0, 1% NP-40, 250 mM LiCl, 1 mM EDTA), and TE buffer (10 mM Tris-HCl pH 8.0, 1 mM EDTA). A total protein-DNA complex was eluted from the beads in 200 μL of elution buffer (TE buffer containing 1% SDS, 100 mM NaCl, 5 mM DTT), treated with RNase A (1 µg/elution, 37 °C, 1 h) and Proteinase K (15 µg/elution, 37 °C, 2 h), reverse-crosslinked (65 °C overnight), and purified using a QIAquick PCR purification kit according to the manufacturer’s instructions (Qiagen, Germantown, MD, USA). Quantitative PCRs were performed with indicated primers. For all ChIP assays, ChIP with histone H3 and control rabbit IgG antibodies were included as a positive control and negative control, respectively. qPCR data from each ChIP with specific antibodies were calculated as percent to input DNA. The antibodies used for ChIP were Normal Rabbit IgG (CST, 2729S), Anti-H3 (Abcam, ab1791), Anti-H3ac (Abcam, ab47915), Anti-H3K4me3 (CST, 9751), and Anti-PolII (Santa Cruz, sc-55492). 

### 4.7. Quantitative PCR (qPCR)

Quantitative PCR (qPCR) was performed in the ABI QuantStudio 5 using Fast SYBR™ Green Master Mix (ABI, Cat# 4385610). The PCR cycle program was as follows: (1) 50 °C for 2 min, 1 cycle; (2) 95 °C for 10 min, 1 cycle; (3) 95 °C for 15 s → 60 °C for 30 s → 72 °C for 30 s, 40 cycles; (4) 72 °C for 10 min, 1 cycle. The primers for the qPCR were as follows: 2-LTR (5′-AACTAGGGAACCCACTGCTTAAG-3′, 5′-TCCACAGATCAAGGATATCTTGTC-3′); LTR (5′-TGTGTGCCCGTCTGTTGTG-3′, 5′- GAGTCCTGCGTCGAGA-3′); Nuc1 (5′-GAGTGCTCAAAGTAGTGTGTGC-3′, 5′- TCTCCTCTGGCTTTACTTTCGC-3′).

### 4.8. CRISPR-Mediated Gene Knockout

Three sgRNAs (AGAGCGTGCAGATAATGACA, AGCTGGCAGACCACAAACTG, ATTCTTCATACCAGGACCAG) targeting PTEN were cloned into pLentiCRISPRv2GFP (ADDGENE #82416). Hela cells were transfected with a pool of three plasmids using lipofectamine 2000. One day post-transfection, the brightest 1% of the GFP-positive cells were sorted by FACS. Single cells from the resulting pool cells were seeded in a 96-well plate and specific gene knockout clones were screened by Western blotting using specific antibodies. 

To rescue the expression of PTEN in PTEN KO cells, KO cells were transduced with pLvx-EF1-IRES-Neo-PTEN (wild type or with indicated mutation) and selected in 800 µg/mL G418 for two weeks. 

### 4.9. siRNA Transfection

A total of 10^5^ HeLa cells were seeded in 6-well plates. Twenty-four hours later, siRNAs were transfected into cells by lipofectamine RNAiMax (Invitrogen) according to the manufacturer’s protocol. After another 24 h, the same siRNA transfection was performed for the second time. Six hours after the second transfection, the siRNA-transfected cells were infected with the virus for further experiments. siRNAs were ordered from CST: control siRNA (CST, 6568S), NF-κB RELA siRNA (CST, 6534S), c-Jun siRNA (CST, 6204S), and SP1 siRNA (CST, 12106S).

### 4.10. Western Blot

For immunoblotting, cells were lysed in a passive lysis buffer (Promega, Cat# E1941) for 10 min. The lysate was clarified by centrifugation at 4 °C for 15 min at 12,000 rpm. The samples were heated at 95 °C in SDS sample buffer and resolved by SDS-PAGE electrophoresis, transferred to a PVDF membrane, and probed with specific antibodies. The antibodies used for the Western blot were as follows: PTEN (Proteintech, 22034-1-AP); Tubulin (Proteintech, 66031-1-Ig); Akt (CST, 4691S); phosphor-Akt (CST, 4060S); RELA (CST, 6956S); SP1 (CST, 9389S); JUN (CST, 9165S).

### 4.11. Quantification and Statistical Analysis

All data presented are mean ± s.d. from three independent experiments. *p* values are from paired two-sided Student *t*-tests.

## Figures and Tables

**Figure 1 viruses-16-00291-f001:**
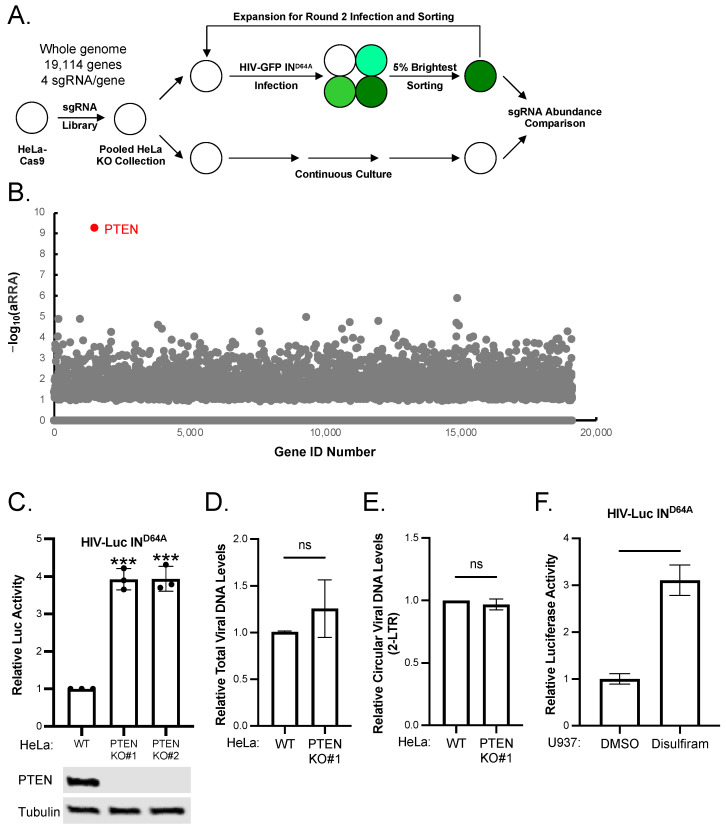
Genome-wide CRISPR knockout screening to identify host factors responsible for the silencing of unintegrated HIV-1 DNA. (**A**) Flowchart of the genome-wide CRISPR knockout screening strategy. (**B**) Dot blots illustrating hits from the screening. The top candidate PTEN was illustrated in red. (**C**) Indicated HeLa cell lines were infected with integrase-deficient (IN^D64A^) HIV-Luc reporter virus. Luciferase activities were measured 48 h post-infection and luciferase activity in WT HeLa cells was set as 1 (top panels). The expression of indicated proteins was determined by Western blot (bottom panels). (**D**,**E**) Indicated HeLa cell lines were infected with integrase-deficient (IN^D64A^) HIV-Luc reporter virus. Forty-eight hours post-infection, total viral DNA (**D**) and circular viral DNA (**E**) were measured by qPCR. The viral DNA levels in WT HeLa cells were set as 1. (**F**) U937 cells were infected with integrase-deficient (IN^D64A^) HIV-Luc reporter virus and treated with DMSO or Disulfiram (1 μM). Luciferase activities were measured 48 h post-infection and luciferase activity in DMSO-treated cells was set as 1. Data are mean ±  s.d.; *n*  =  3 independent experiments. ns: *p* > 0.05; ***: *p* < 0.001.

**Figure 2 viruses-16-00291-f002:**
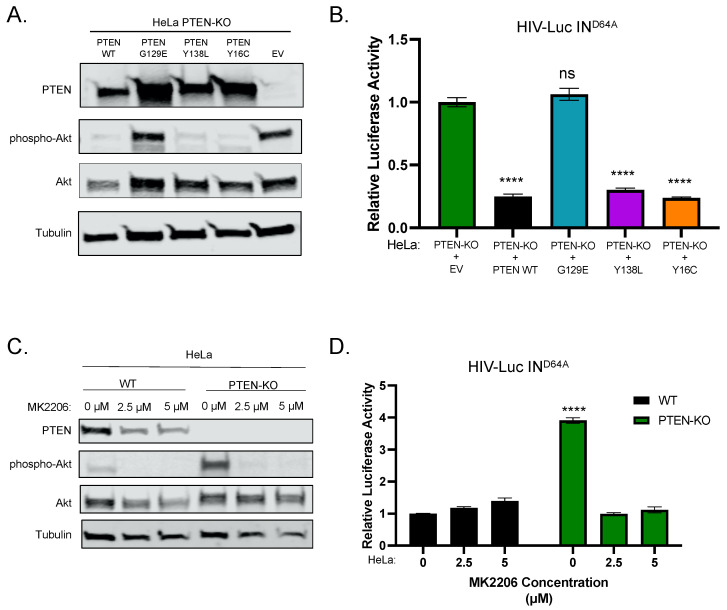
PTEN’s lipid phosphatase activity is essential for the silencing of unintegrated HIV-1 DNA. (**A**,**B**) PTEN-KO HeLa cells were transduced with empty vector (EV), wild-type PTEN (PTEN WT), or indicated mutant forms of PTEN. Cells were infected with integrase-deficient (IN^D64A^) HIV-Luc reporter virus. The expression of indicated proteins was determined by Western blot (**A**). Luciferase activities were measured 48 h post-infection and luciferase activity in PTEN-KO (+EV) cells was set as 1 (**B**). (**C**,**D**) Parental (WT) and PTEN-knockout (PTEN-KO) HeLa cells were treated with MK2206. Cells were then infected with integrase-deficient (IN^D64A^) HIV-Luc reporter virus. The expression of indicated proteins was determined by Western blot (**C**). Luciferase activities were measured 48 h post-infection and luciferase activity HeLa WT cells was set as 1 (**D**). Data are mean ±  s.d.; *n*  =  3 independent experiments. ns: *p* > 0.05; ****: *p* < 0.0001.

**Figure 3 viruses-16-00291-f003:**
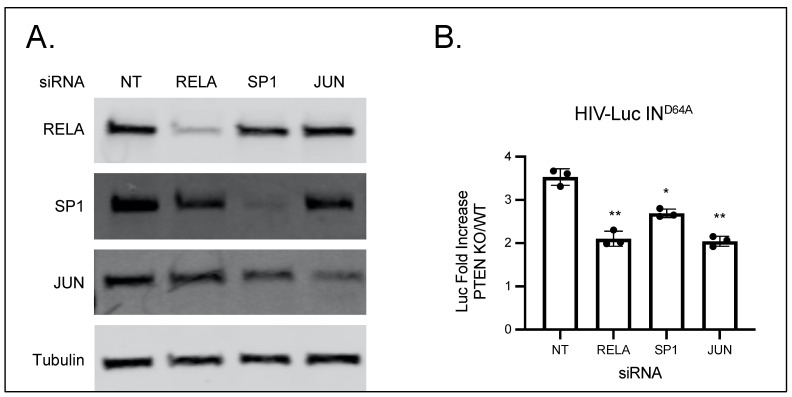
Transcription factors essential for the activation of unintegrated HIV-1 DNA in PTEN-KO cells. (**A**) HeLa cells were transfected with indicated siRNA, and the expression of indicated proteins was detected by Western blot to determine the siRNA knockdown efficiency. (**B**) HeLa WT and PTEN-KO cells were transfected with indicated siRNA and then infected with integrase-deficient (IN^D64A^) HIV-Luc reporter virus. Luciferase activities were measured 48 h post-infection and fold increase (PTEN KO/WT) was calculated as the ratio of luciferase activities in KO cells to activities in WT cells. Data are mean ±  s.d.; *n*  =  3 independent experiments. *: *p* < 0.05; **: *p* < 0.01.

**Figure 4 viruses-16-00291-f004:**
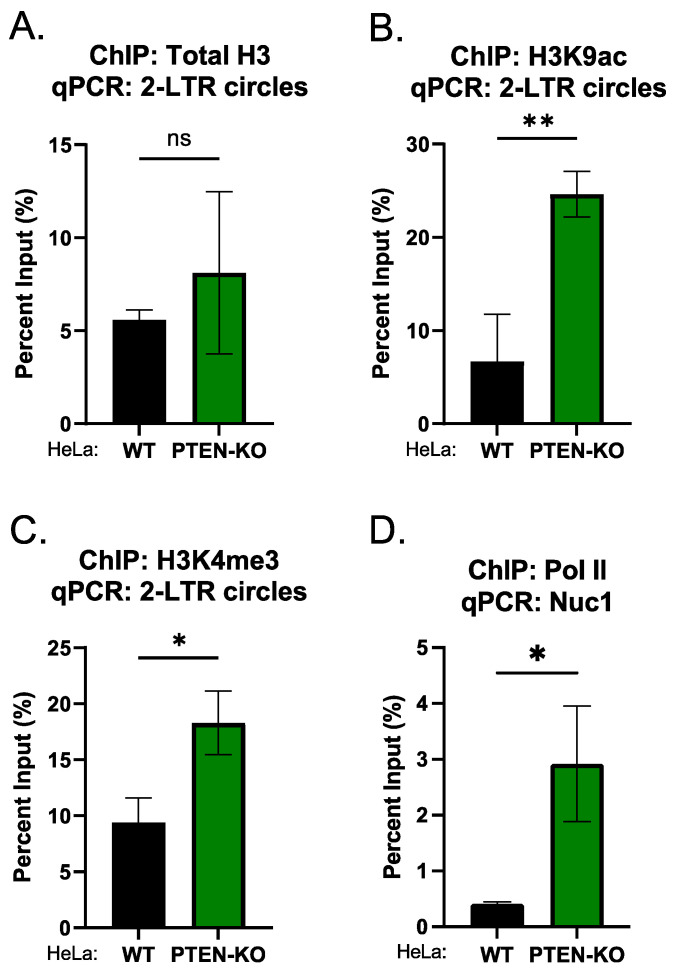
Knockout of PTEN increased the levels of active epigenetic marks on unintegrated HIV-1 DNA. Indicated HeLa cell lines were infected with integrase-deficient (IN^D64A^) HIV-Luc reporter virus. Forty-eight hours post-infection, ChIP was performed using antibodies to H3 (**A**), H3K9ac (**B**), H3K4me3 (**C**), and Pol II (**D**), followed by qPCR using primers targeting 2-LTR (**A**–**C**) or Nuc1. qPCR data from each ChIP were calculated as percent of input DNA. Data presented are mean ± s.d. from three independent experiments (*n* = 3). ns: *p* > 0.05; *: *p* < 0.05; **: *p* < 0.01.

## Data Availability

This paper does not report original dataset or code.

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
