# Peer review of "PTEN Mediates the Silencing of Unintegrated HIV-1 DNA"

_viruses, 2024, doi:10.3390/v16020291_

Round 1

Reviewer 1 Report

Comments and Suggestions for Authors

In this short yet elegant report, Phan et al. describe a CRISPR screen of host factors important for transcriptional silencing of unintegrated HIV-1 in HeLa cells, and report the discovery and validation of PTEN as a host repression factor of unintegrated HIV-1 DNA. PTEN silencing of unintegrated HIV-1 involves the dephosphorylation of PIP3, which prevents the activation of Akt and the downstream transcription factors NFkB, SP1 and JUN. Evidance of how activation of Akt is linked to NFkB activation and deposition of euchromatic histone marks on unintegrated viral DNA is rather lacking, yet given the scope of the story and the journal, this reviewer does not think these are required for publication. Another suggestion for improvement (again not required) is the usage of disulfiram to validate that PTEN is also important for suppression of D64A-mutant integrase HIV-GFP in PTEN-expression CD4+ T cells, a quick way to patch up the lack of direct evidance that PTEN is important for HIV-1 suppression outside of HeLa cells.

Issue with Fig. 4: panels were not labeled A, B, C, yet Figs. 4A, 4B, 4C were mentioned in the text.

Comments on the Quality of English Language

The writing is overall fine, with the exception of a few typos:
- page 1, line 43: LTR miswritten as LRT
- page 2, line 55: associateD
- page 2, line 80: algorithm misspelled.

Reviewer 2 Report

Comments and Suggestions for Authors

Comments for the Authors

virus-2810502

PTEN Mediates the Silencing of Unintegrated HIV-1 DNA

Authors: An Thanh Phan, and Yiping Zhu

In this study, the authors use genome-wide CRISPR knockout screening to identify a novel host factor, PTEN,  involved in suppressing the transcription of unintegrated HIV-1 DNAs. The authors further establish stable cell lines expressing either wildtype and mutated (G129E, Y138L, and Y16C) PTEN in PTEN-depleted HeLa cells and infect them with integrase-deficient (IND64A) HIV-Luc reporter virus. The authors claim that mutation G129E associated with lipid dephosphorylation and deactivation of Akt is pivotal for silencing unintegrated HIV-1 DNA. Cellular transcription factors (NF-kB, Sp1, and AP-1), which are involved in the downstream Akt signaling pathway are also shown to interfere with the transcription of unintegrated HIV-1 DNA using luciferase reporter assay. Finally, the authors assay and identify histone modifications (H3K9ac and H3K4me3), which participate in the remodeling of the conformation of HIV-1 5’LTR during the silencing process governed by PTEN.

Overall, this study is straightforward and unveils the potential function of PTEN in silencing the transcription of unintegrated HIV-1 DNA. There are still a few issues that the authors may try to clarify, as suggested below:

1. The authors have discussed several identified host factors, e.g. SMC5/6 involved in silencing unintegrated HIV-1 DNA. Please also briefly discuss another recently identified host factor, POLE3 (Thenin-Houssier et al., 2023, DOI:10.1126/sciadv.adh3642), which also plays the role of silencing and escape from immune sensing.

2. Two factors, PTEN and ZFR, were released from the CRISPR screening. It is not clear to me the reason the authors only focus on PTEN rather than ZFR.

3. It is not clear to me, as compared to the experimental condition PTEN-KO + EV (Figure 2B), the fold change shows four times enrichment in the condition using PTEN-KO (Figure 2D) in the absence of MK2206 inhibitor. 

4. Please clarify the difference between PTEN-KO and PTEN-deficient cells described in this study. It would make the readers easier to follow the article as the authors could keep the consistency of the terms described for the identical materials. 

5. Western blot (Figure 3A) indicated that the efficiency of siRNA knockdown on RELA, Sp1, and JUN, especially JUN should be further optimized. The authors should consider using CRISPR to knock out the genes encoding incident proteins.

6. The authors examined the histone loading (H3) on unintegrated HIV-1 DNA using ChIP qPCR. It is suggested to also test RNA PolII loading on the LTR of unintegrated HIV-1 DNA.

Reviewer 3 Report

Comments and Suggestions for Authors

The mechanisms of regulation of unintegrated HIV-1 DNA are poorly understood, though a number of host factors have been identified which participate in this regulation. Previous observations suggest that the underlying regulatory mechanisms may be epigenetic in nature. Here Phan and Zhu performed a genome-wide CRISPR knockout screening, identifying host genes which relieved the silencing of unintegrated HIV-1 DNA when knocked out. The results of their screen revealed PTEN and ZFR as candidates for involvement in these mediatory processes. This work is interesting however several points should be addressed before this manuscript is ready for publication.

 While it’s almost certain that the viral DNA is unintegrated in the absence of functional integrase, it’s important to rule out other possible causes for the observations. For example, is it possible that in the presence of PTEN with lipid phosphatase activity, the genome is excluded from the nucleus, or fails to uncoat? This could be easily addressed by showing that similar levels of 2-LTR circles are present under the conditions being compared. 2-LTR circles are already being quantitated in Figure 4, showing that the methodology is readily available.

 Importantly, there is no mention of statistical analysis. First the data are described as being the mean +/- SD. Are the error bars showing +/- one standard deviation? The significance of the asterisks is not defined nor are the comparisons being made. Further no statistical test is mentioned.

 For the CRISPR/Cas9 KO lines it’s stated that two clones are generated for both PTEN and ZFR. It’s not clear whether these were generated in wild-type HeLa cells or the Cas9-expressing line or whether two different guide RNAs are being used. Further no gRNA sequences were provided. This is of course important for others seeking to replicate this work.

 PTEN and ZFR are described as standing out, however no criteria were defined for their selection. Also, no explanation was provided for dropping the analysis of ZFR. Finally, were any other genes found that could/should be expected?

 In Figure 2, MK2206 appears to reduce PTEN levels in WT cells. This should be addressed.

 In previous work, AKT inhibitors were shown to have anti-viral effects in macrophages (Akt inhibitors as an HIV-1 infected macrophage-specific anti-viral therapy Pauline Chugh, Birgit Bradel-Tretheway, Carlos MR Monteiro-Filho, Vicente Planelles, Sanjay B Maggirwar, Stephen Dewhurst & Baek Kim). It would be interesting to mention possible similarities and differences.

 The text references Figs. 4A, B and C, but Fig. 4 is not labeled as such.

Comments on the Quality of English Language

Generally fine. There are a few typos and some minor issues with wording.

Round 2

Reviewer 2 Report

Comments and Suggestions for Authors

The authors have answered my comments point by point. I do not have further questions.

Reviewer 3 Report

Comments and Suggestions for Authors

The revisions are satisfactory.